# Multi-City Analysis of the Acute Effect of Polish Smog on Cause-Specific Mortality (EP-PARTICLES Study)

**DOI:** 10.3390/ijerph20085566

**Published:** 2023-04-18

**Authors:** Michał Święczkowski, Sławomir Dobrzycki, Łukasz Kuźma

**Affiliations:** Department of Invasive Cardiology, Medical University of Bialystok, 15-089 Bialystok, Poland

**Keywords:** air pollution, polish smog, particulate matter, nitrogen dioxide, acute coronary syndrome, ischemic stroke, mortality

## Abstract

Polish smog is a specific type of air pollution present in Eastern Poland, which may cause particularly adverse cardiovascular effects. It is characterized primarily by high concentrations of particulate matter (PM) and different favorable conditions of formation. Our study aimed to assess whether PM and nitrogen dioxide (NO_2_) have a short-term impact on mortality due to acute coronary syndrome (ACS) and ischemic stroke (IS). The study covered the years 2016–2020, a total of 6 million person-years from five main cities in Eastern Poland. To evaluate the association between air pollution and cause-specific mortality, a case-crossover study design with conditional logistic regression was used at days with LAG from 0 to 2. We recorded 87,990 all-cause deaths, including 9688 and 3776 deaths due to ACS and IS, respectively. A 10 μg/m^3^ increase in air pollutants was associated with an increase in mortality due to ACS (PM_2.5_ OR = 1.029, 95%CI 1.011–1.047, *p* = 0.002; PM_10_ OR = 1.015, 95%CI 1–1.029, *p* = 0.049) on LAG 0. On LAG 1 we recorded an increase in both IS (PM_2.5_ OR = 1.03, 95%CI 1.001–1.058, *p* = 0.04) and ACS (PM_2.5_ OR = 1.028, 95%CI 1.01–1.047, *p* = 0.003; PM_10_ OR = 1.026, 95%CI 1.011–1.041, *p* = 0.001; NO_2_ OR = 1.036, 95%CI 1.003–1.07, *p* = 0.04). There was a strong association between air pollution and cause-specific mortality in women (ACS: PM_2.5_ OR = 1.032, 95%CI 1.006–1.058, *p* = 0.01; PM_10_ OR = 1.028, 95%CI 1.008–1.05, *p* = 0.01) and elderly (ACS: PM_2.5_ OR = 1.03, 95%CI 1.01–1.05, *p* = 0.003; PM_10_ OR = 1.027, 95% CI 1.011–1.043, *p* < 0.001 and IS: PM_2.5_ OR = 1.037, 95%CI 1.007–1.069, *p* = 0.01; PM_10_ OR = 1.025, 95%CI 1.001–1.05, *p* = 0.04). The negative influence of PMs was observed on mortality due to ACS and IS. NO_2_ was associated with only ACS-related mortality. The most vulnerable subgroups were women and the elderly.

## 1. Introduction

Cardiovascular disease (CVD) is the most common cause of death responsible for almost 18 million deaths worldwide, with ischemic heart disease and stroke accounting for 9 and 6 million, respectively [1]. Simultaneously, air pollution has become one of the largest health and environmental problems. According to the Global Burden of Disease 2019, approximately 6.7 million deaths annually are attributed to air pollution [2]. Currently, the loss of life expectancy due to poor air quality exceeds even that of tobacco smoking, which reflects the scale of the problem [3].

Most of the studies on this topic have been performed in highly polluted areas in Asia, mostly China. However, these results cannot be easily extrapolated to other regions since there are substantial differences in smog composition due to contrasting weather conditions and other sources of emission. Morbidity and mortality bound to indoor and outdoor air pollution are also dissimilar in Europe and Asia. Furthermore, smog composition changes over time, as the average surface temperature and height of the inversion layer keep increasing [4]. Even though China’s clear air actions have resulted in a 15% and 59% decrease in premature deaths attributable to short and long-term exposure to PM_2.5_, respectively, ozone-bound mortality has increased continuously during the study period and the health costs of poor air quality have remained relatively high [5].

The harmful effect of long-term exposure to air pollution on atherothrombotic CVD morbidity and mortality has been recognized in the last decades [6,7,8]. Nevertheless, some studies, especially from Europe have presented ambiguous results regarding the short-term influence of poor air quality. Byrne et al., revealed no significant impact of air pollution on ischemic stroke (IS) onset without division into seasons, whereas O’Donnell et al., presented that an increase in PM_2.5_ concentration was not associated with the risk of IS in the overall population [9,10]. In a large cohort of postmenopausal women, there was no association between five analyzed air pollutants and IS occurrence [11]. Moreover, in a study performed in Poland, it was demonstrated that carbon monoxide (CO) might have a protective effect on IS onset, whereas other air pollutants increased IS-related morbidity on different days after exposure depending on age and sex [12]. Other unclear matters remaining in the available literature are lag patterns and vulnerability to smog in specific subgroups. Collart et al., presented interesting results, in which younger people were similarly sensitive to air pollution as the elderly, but the negative effect was more delayed for them [13]. On the other hand, in line with most of the available research, Li et al., found a stronger association between smog and the incidence of myocardial infarction in older individuals [14]. The effect of air pollution also differs between sexes, lifestyles, socioeconomic groups, ethnicity, and comorbidities [15,16]. Some studies have suggested that smog and its effect might even vary between urban and rural areas in the same country [17]. Moreover, as Kim et al., revealed, females were far more susceptible to PM_10_ long-term exposure in South Korea—a country with a moderate concentration of air pollution [15]. Similar to the Gaines et al., study, where women, the elderly, and people with comorbidities were more sensitive to poor air quality [16].

Eastern Poland is an area with an unusual type of air pollution, that significantly differs from the London smog and Los Angeles smog. The so-called Polish smog is composed mainly of particulate matter (PM) and polycyclic aromatic hydrocarbons, often not accompanied by high sulfur dioxide (SO_2_) and CO concentrations. Its formation is favored by high atmospheric pressure and low temperature, especially in winter when heat demand increases causing higher emissions of air pollution from household heating systems [18,19]. The adverse effect of Polish smog has been well documented in one of the analyzed cities, as it increased hospitalization rates due to acute coronary syndromes (ACS) and IS [12,20]. However, the influence of air pollution in the other four cities has never been analyzed. Furthermore, Poland was one of the three countries with the highest PM_2.5_-bound premature mortality according to a study performed by Khomenko et al., [21]. We argue that Polish smog, which is present in the analyzed region, may be very dangerous to public health.

The main sources of air pollution in the analyzed area are low emissions due to the combustion of poor-quality solid fuels in low-efficiency furnaces and traffic. Exposure to air pollutants promotes inflammation, intracellular oxidative stress, endothelial dysfunction, and genetic changes, which all combined have a negative effect on the cardiovascular system [22,23].

This study aimed to assess whether Polish smog has a short-term influence on mortality due to ACS and IS. Additionally, we wanted to identify lag patterns of an acute effect of exposure to air pollution and the most vulnerable groups.

## 2. Material and Methods

### 2.1. Study Design

Retrospective analyses were conducted. Data on mortality were obtained from Central Statistical Office in Poland.

We analyzed all deaths in 5 voivodeships located in the eastern part of Poland (Lubelskie, Podkarpackie, Podlaskie, Swietokrzyskie, and Warminsko-Mazurskie) in the years 2016–2020 (*n* = 432,544) (Figure 1 and Figure 2). For further analysis, we included all deaths recorded in 5 voivodeships capitals—Bialystok, Kielce, Lublin, Olsztyn, and Rzeszow (*n* = 87,990). Non-CVD-related deaths were excluded from the study. In our detailed analysis, we evaluated whether air quality affected mortality due to ACS and IS up to 2 days after exposure in the aforementioned 5 biggest cities in the eastern part of Poland.

A case-crossover (CCO) study was used to assess the effect of particulate matter (PM) concentrations, NO_2_, and weather conditions on mortality. Each of the pollutants was modeled individually.

The death day was defined as the case period, while the control periods included all days from the same week in the same month as the case period. This method was considered to adjust for the effects of long-term trends, seasonality, and the days of the week, using 3–4 days as controls for each case and allowed to provide implicit control of all the known and unknown confounders that are unlikely to vary nonrandomly during the index and reference periods.

### 2.2. Studied Region

Our studied area is located in the eastern part of Poland and is inhabited by over 8 million people (Figure 2). It contains 5 voivodeships and is often described as a region with a high share of agriculture and low industrialization. Table 1 presents the characteristics of this region’s further analyzed 5 voivodeship capitals. Almost 1.2 million people combined live in these aforementioned cities. Differences between the studied cities in population density, femininity ratio, and share of elderly people are noticeable.

#### Pollution and Meteorological Data

The data on air pollution and gases were obtained from the Voivodeship Inspectorate for Environmental Protection. In the analysis, we used the concentration of particulate matter with a diameter of 2.5 μm or less (PM_2.5_) or 10 μm or less (PM_10_), and nitrogen dioxide (NO_2_).

In Białystok, the data on the concentrations of NO_2_ and PM_2.5_ were obtained from the station with international code—PL0496A (GPS—53°13′ N, 23°18′ E). PM_10_ measurements were obtained from station PL0148A (GPS—53°13′ N, 23°16′ E). In Kielce, we collected data on all 3 pollutants from 2 stations: from 01/01/2016 to 05/24/2018—PL0283A (GPS—50°87′ N, 20°60′ E) and from 07/01/2018 to 12/31/2020—PL0704A (GPS—50°88′ N, 20°63′ E). In Lublin, the data on NO_2_, PM_2.5_, and PM_10_ concentrations were obtained from station PL0507A (GPS—51°26′ N, 22°57′ E). In Olsztyn, the measurements of 3 analyzed pollutants were collected from station: PL0175A (GPS—53°79′ N, 20°49′ E). In Rzeszow, we obtained data on concentrations of NO_2_ and PM_10_ from station PL0495A (GPS—50°02′ N, 22°01′ E). PM_2.5_ concentrations were acquired from 2 stations: from 01/01/2016 to 31/12/2018—PL0495A (GPS—50°02′ N, 22°01′ E) and from 1 January 2019 to 31 December 2020—PL0736A (GPS—50°04′ N, 22°00′ E). In the analysis, the exceedance of air pollution norms was determined based on the Air Quality Guidelines (AQG) 2021 [24]. The 24-h concentrations recommended by the WHO are 45 µg/m^3^, 15 µg/m^3^, and 25 µg/m^3^ for PM_10_, PM_2.5_, and NO_2_, respectively [24].

The daily meteorological data, including mean temperature, the daily level of relative humidity, and mean atmospheric pressure were obtained from the Institute of Meteorology and Water Management. There is a station recording meteorological data in each of the 5 analyzed cities: Bialystok—station identification number (ID) 353230295, Kielce—ID 350200570, Lublin—ID 251160190, Olsztyn—353200272, Rzeszow—ID 350220580.

The study material lacked about 1.86% of daily data on air pollution. No data were lost on meteorological conditions.

### 2.3. Statistical Analysis

The distribution of variables was evaluated using the Shapiro–Wilk test. Categorical variables are expressed as numbers and percentages, whereas continuous variables are presented as mean values with standard deviation (SD). Kruskal–Wallis and analysis of variance test were used for comparative analysis.

The association between air pollution and the occurrence of CVD deaths was estimated by odds ratios (ORs) with 95% confidence intervals (CIs) using conditional logistic regression (CLR) at days with LAG from 0 to 2. The average temperature, atmospheric pressure, and relative humidity were used as covariates in the CLR model. We used a natural cubic spline with 4 degrees of freedom for the weather conditions–mortality function. The association between weather conditions and the occurrence of ACS- and IS-related deaths was estimated by OR with 95% CIs using CLR at days with LAG from 0 to 2. Differences between both sexes and age groups (under and over 65 years old) are presented as relative risks (RR) with 95% CIs. The threshold of statistical significance for all tests was set at *p*-value (*p*) < 0.05. Periods without data were excluded from the analysis.

All analyses were performed using MS Excel (Microsoft, 2020, version 16.40, Redmond, WA, USA) and XL Stat (Addinsoft, 2020, version 2020.03.01, New York, NY, USA). The study protocol conformed to the ethical guidelines of the 1975 Declaration of Helsinki and was approved by the Bioethics Committee of the Medical University of Bialystok (approval number APK.002.81.2022).

The study was financed from the funds of the National Science Center granted under the contract number UMO-2021/41/B/NZ7/03716 and the funds of the Medical University of Bialystok granted under the contract number B.SUB.23.101 and B.SUB.23.509. The study was registered at ClinicalTrials.gov (NCT05198492).

## 3. Results

We recorded 432,544 all-cause deaths in the years 2016–2020 in five analyzed voivodeships. A total of 87,990 all-cause deaths were noted in five voivodeships capitals. Inclusion criteria for this study were met for 34,907 deaths due to CVD, including 9688 and 3776 deaths related to ACS and IS, respectively. In the ACS group, 51.11% were male with a mean age of 78.06 years old (SD = 11.32). Out of IS-related deaths, 43.43% were male, and the mean age was 79.13 years old (SD = 11.04).

The highest ACS-related and IS-related standardized death rates (SDR) were recorded in Bialystok, while the lowest ACS-related and IS-related SDR were noted in Olsztyn and Rzeszow, respectively (ACS: 241.9 per 100,000 population/year vs. 114.56 per 100,000 population/year; *p* < 0.001 and IS: 86.79 per 100,000 population/year vs. 63.61 per 100,000 population/year; *p* < 0.001). The detailed mortality statistics regarding the five analyzed cities are depicted in Table 2.

In Figure 3, SDRs related to all-cause deaths (Figure 3A), CVD (Figure 3B), ACS (Figure 3C), and IS (Figure 3D) are presented. The highest CVD-related SDR was recorded in Kielce in 2020 (872.1 per 100,000 population/year), while the lowest was noted in Olsztyn in 2018 (471.2 per 100,000 population/year). The highest ACS-related SDR was observed in Kielce in 2020 (357 per 100,000 population/year), whereas the lowest was recorded in Lublin in 2019 (94.54 per 100,000 population/year). The highest IS-related SDR was noted in Bialystok in 2017 (96.05 per 100,000 population/year), while the lowest was in Rzeszow in 2018 (35.22 per 100,000 population/year).

Taking into account the seasonal variation in the frequency of deaths related to ACS and IS, we recorded the highest mean/day in the winter ((ACS: *n* = 1.103/day (SD = 1.29) and IS: *n* = 0.439 (SD = 0.689)), while the lowest in the summer ((ACS: N = 0.97/day (SD = 1.14) and IS: *n* = 0.375 (SD = 0.625)), *p* = 0.006 and *p* = 0.005, respectively, for pairwise comparison. Regarding the incidence of death due to ACS throughout the week, the highest mean/day was observed on Mondays (*n* = 1.142/day (SD = 1.293)) and the lowest was observed on the weekends (Friday: *n* = 1.027/day; *p* = 0.045, Saturday: *n* = 1.012/day; *p* = 0.02, Sunday: *n* = 1.033/day; *p* = 0.02). On the other hand, the highest occurrence of IS-related deaths was noted on Tuesdays (*n* = 0.442/day), while the lowest was noted on Sundays (*n* = 0.369/day); *p* = 0.002 for pairwise comparison. See Figure 4.

For detailed statistics concerning air pollution concentrations in individual years or by seasons and share of days with exceeded daily means according to AQG 2021, please refer to Table 3. The highest share of days with an exceeded daily mean of PM_2.5_ was observed in Lublin (*n* = 1054 (58.92%)), while the lowest was in Bialystok (*n* = 666 (37.46%)). For PM_10_ the highest share of days with exceeded daily mean concentration was recorded in Kielce (*n* = 347 (19.54%)), whereas the lowest was in Olsztyn (*n* = 116 (6.49%)). The highest share of days with exceeded daily mean levels of NO_2_ was noted in Kielce (*n* = 740 (41.39%)) and the lowest was observed in Bialystok (*n* = 69 (3.83%)).

### 3.1. Cohort Analyses

The 10 μg/m^3^ increase in PM_2.5_ (OR = 1.029, 95% CI 1.011–1.047, *p* = 0.002) and PM_10_ (OR = 1.015, 95% CI 1–1.029, *p* = 0.049) concentrations were associated with an increase in mortality due to ACS on the day of the exposure in the overall population. This effect was also observed on LAG 1 (PM_2.5_: OR = 1.028, 95% CI 1.01–1.047, *p* = 0.003; PM_10_: OR = 1.026, 95% CI 1.011, *p* = 0.001). Additionally, NO_2_ also caused increased mortality due to ACS on LAG 1 (OR = 1.036, 95% CI 1.003–1.07, *p* = 0.04). On LAG 2 only PM_10_ had a significant impact on ACS-related mortality (OR = 1.022, 95% CI 1.001–1.044, *p* = 0.04). An increase of 10 μg/m^3^ in PM_2.5_ caused an increase in the number of deaths due to IS on LAG 1 (OR = 1.03, 95% CI 1.001–1.058, *p* = 0.04) (Figure 5).

### 3.2. Subgroup Analyses

The impact of PM_2.5_ on the increased mortality due to ACS was observed on the day of exposure (OR_Women_ = 1.033, 95% CI 1.007–1.06, *p* = 0.01) and LAG 1 (OR_Women_ = 1.032, 95% CI 1.006–1.058, *p* = 0.01). PM_10_ had a significant impact on ACS-related mortality on the next day after exposure in men (OR_Men_ = 1.023, 95% CI 1.002–1.044, *p* = 0.03) and women (OR_Women_ = 1.028, 95% CI 1.008–1.05, *p* = 0.01). Moreover, a 10 μg/m^3^ increase in PM_10_ concentration was associated with an increase in the number of deaths due to IS on LAG 0 (OR_Women_ = 1.032, 95% CI 1.002–1.063, *p* = 0.038) in females. There was no significant impact of air pollution on IS-related mortality in the male population (Figure 6). The association between exposure to PMs and ACS-related mortality was larger among women on LAG 0 (PM_2.5_: RR = 1.024; 95% CI 1.010–1.038; *p* < 0.001) and LAG 1 (PM_2.5_: RR = 1.025; 95% CI 1.011–1.043; *p* < 0.001 and PM_10_: RR = 1.006; 95% CI 1.001–1.011; *p* < 0.001) compared to men. The association between exposure to PM_10_ and the number of deaths due to IS on LAG 0 was also greater in females (RR = 1.006; 95% CI 1.012–1.015; *p* < 0.001).

The effect of PM_2.5_ on ACS-related mortality was noted on LAG 0 in the elderly (OR_>65y.o_ = 1.029, 95% CI 1.009–1.048, *p* = 0.004). PM_2.5_ and PM_10_ had a significant impact on both ACS (PM_2.5_: OR_>65y.o_ = 1.03, 95% CI 1.01–1.05, *p* = 0.003; PM_10_: OR_>65y.o_ = 1.027, 95% CI 1.011–1.043, *p* = 0.001) and IS (PM_2.5_: OR_>65y.o_ = 1.037, 95% CI 1.007–1.069, *p* = 0.014; PM_10_: OR_>65y.o_ = 1.025, 95% CI 1.001–1.05, *p* = 0.0042) mortality on LAG 1. Furthermore, we observed an association between an increase in NO_2_ concentration and an increase in mortality due to ACS on the next day after exposure in people under (OR_<65y.o_ = 1.047, 95% CI 1.011–1.084, *p* = 0.011) and over (OR_>65y.o_ = 1.047, 95% CI 1.011–1.084, *p* = 0.011) 65 years old (Figure 7). On LAG 1 the associations between exposure to PMs and occurrence of ACS (PM_2.5_: RR = 1.019; 95% CI 1.002–1.024; *p* < 0.001; PM_10_: RR = 1.013; 95% CI 1.008–1.017; *p* < 0.001) and IS (PM_2.5_: RR = 1.038; 95% CI 1.033–1.043; *p* < 0.001; PM_10_: RR = 1.044; 95% CI 1.038–1.049; *p* < 0.001) were greater in elderly.

In the Appendix A, we present the influence of temperature, relative humidity, and atmospheric pressure on the occurrence of death due to ACS and IS. There was no statistically significant impact of the aforementioned factors on ACS- and IS-related mortality.

## 4. Discussion

The findings of our study indicate that environmental factors such as air quality have a significant impact on mortality due to major cardiovascular diseases. Depending on age and sex, some subgroups may be more vulnerable to smog than others. Additionally, we recorded chronobiological trends in mortality due to ACS and IS. To our knowledge, this is the biggest study of its kind performed in this region.

In our study, the harmful effect of air pollutants on the general population was more pronounced in the ACS group, especially the day after exposure (Figure 5). Liu et al., presented an association between a 10 μg/m^3^ increase in air pollutants (PM_2.5_, PM_10_, NO_2_) and myocardial infarction (MI) mortality on LAG 0 and LAG 1 in the overall population, which comes in line with our results [25]. Additionally, a study conducted in the region with relatively low levels of air pollutants by Gestro et al., showed that exposure to PM_2.5_ was a risk factor for ACS incidence, particularly in elderly people [26]. Short-term exposure to all three analyzed air pollutants was related to greater incidence and mortality due to STEMI in Barcelona, Spain [27]. On the other hand, Vaudrey et al., demonstrated no independent impact of PM_2.5_, PM_10,_ and NO_2_ on the onset of MI [28]. Bateson and Schwartz and Ueda et al., presented a positive correlation between short-term exposure and PMs and MI-related mortality, but their results were not statistically significant [29,30]. It is suspected that exposure to PM_2.5_ can trigger ACS only in patients with preexisting coronary artery disease [31]. Furthermore, Berglind et al., suggested that air pollution has a stronger effect on MI survivors than on the overall population [32].

In our results, only PM_2.5_ had a significant impact on IS-related mortality the day after exposure (Figure 5). There is a severe lack of research on mortality due to IS as studies have mostly focused on its incidence. Franklin et al., showed a 1.03% increase in mortality due to all types of strokes on LAG 1 associated with a 10 μg/m^3^ increase in PM_2.5_ concentration [33]. In a study conducted in Wuhan, China, PM_10_ had a significant impact on an increased risk of fatal stroke on the day of exposure [34]. A similar effect was noted in Moscow, Russia on all cerebrovascular disease (CbVD) mortality and the effect was stronger in the older population [35]. Kettunen et al., demonstrated an influence of PM_2.5_ on stroke mortality only in the warm season [36]. On the other hand, Yorifuji et al., presented no significant impact of PM_2.5_ and NO_2_ on mortality due to IS up to 2 days after exposure [37]. Furthermore, no effect of PM_10_ was noticeable on CbVD mortality in Madrid, in contrast to circulatory and respiratory causes [38]. Some of the most recent studies presented mixed results on gaseous pollutants’ influence on IS incidence. Chen et al., showed that an increase in NO_2_ levels caused greater IS occurrence on LAG 3 and LAG 4, whereas Cui et al., did not find such an association [39,40].

When analyzing age-specific differences in air pollution impact on ACS- and IS-related mortality, a more noticeable effect was present in elderly people (Figure 7). However, there are conflicting results in the available literature. It might seem that due to less prevalent classical risk factors in a younger population, air pollution should be the most prominent. On the other hand, a higher prevalence of coronary artery disease and carotid atherosclerosis in the elderly could magnify the impact of smog. Most of the recent studies have demonstrated a greater incidence of MI due to smog exposure in older adults [25,41,42,43,44]. Although some studies have suggested a greater influence of air pollution on IS incidence in young people, the majority of so far conducted papers are consistent with our results. Yitshak Sade et al., presented not only a significant impact of PM_2.5_ and PM_10_ increase on IS incidence among individuals under 55 years old on the day of the event but also denied the same effect in the elderly [45]. Chen et al., revealed no significant differences between air pollution and stroke hospitalization in both analyzed groups divided by age [46]. On the other hand, Wellenius et al., found a linear PM_10_ exposure–response relationship on stroke mortality in the elderly, while in younger subjects that was not the case [47]. Moreover, the Chang et al., study showed that people over 80 years old were the most susceptible group, considering the impact of PM_2.5_ on hospitalization rates due to stroke [48].

In our study, females were more vulnerable to air pollution than men (Figure 6). First of all, due to sex differences in breathing patterns, the deposition of particles is higher in women’s lungs [49]. Moreover, according to Sørensen et al., study, air pollution causes more oxidative stress in women because of a lower red blood count compared to men [50]. Liang et al., demonstrated a stronger association between PM_2.5_ and acute MI count in the female population [43]. A similar effect for PM_10_ and MI morbidity was found in a study conducted in Tuscany, Italy [44]. Furthermore, a 10 μg/m^3^ increase in the 3-day average exposure to PM_2.5_ was associated with a 36% increase in the 30-day risk of death among female patients originally hospitalized for ischemic heart disease [51]. Chen et al., found a stronger correlation between exposure to four analyzed air pollutants (PM_2.5_, PM_10_, NO_2_, SO_2_) and stroke hospitalization in women compared to men up to 5 days after exposure [46]. Meanwhile, when comparing the influence of particulate matter on transient ischemic attack morbidity, males were a more susceptible group [52].

Variations in impact may also be due to time and physical activity (PA) outdoors. Even though PA has been proven beneficiary to people’s health, exercising outdoors in heavily polluted areas may have a debatable influence on CVD incidence [53,54,55]. ESC Guidelines on CVD prevention suggest that patients at high risk should be encouraged to avoid long-term exposure to air pollution [56]. Submaximal PA changes the breathing pattern from nasal to oral and deepens it, which bypasses natural protective barriers against air pollutants and increases the penetration of particles. Kim et al., showed that an increase in PA in a highly polluted environment may harm younger adults [57]. On the other hand, most of the recent studies have suggested that exercising mitigates the negative effect of air pollution [58,59]. In a population of women in the United States, a regular work out was associated with lower CVD risk and overall mortality irrespectively of PM_2.5_ concentration [58]. In another study performed by Kim et al., moderate to vigorous exercising reduced CVD risk in both low and highly polluted environments [59]. As a protective factor for CVD, PA is very important; however, we argue that cardiologists should educate high-risk patients about the harmful influence of air pollution. These individuals should follow air quality forecasts to plan their outdoor PA around them and avoid exercising near traffic.

As our results showed, mortality due to CVD, and especially ACS, rose during the analyzed period with its peak in 2020 (Figure 3). We argue that this sudden increase may be related to the coronavirus disease 2019 (COVID-19) pandemic. First of all, severe acute respiratory syndrome coronavirus 2 (SARS-CoV-2) infection can transform stable coronary artery disease into ACS by causing a release of pro-coagulant molecules [60]. A study by Modin et al., revealed 5- and 10-times higher incidences of acute MI and IS, respectively, during 14 days after SARS-CoV-2 infection, compared with the control interval [61]. Katsoularis et al., presented a significantly increased occurrence of two aforementioned diseases even up to 4 weeks following COVID-19, when compared to control individuals [62]. Moreover, COVID-19-positive patients were less likely to receive invasive coronary angiography, percutaneous coronary intervention, and dual antiplatelet therapy [63]. Secondly, the beginning of the COVID-19 pandemic in Poland was characterized by the serious disorganization of health care. Closed primary healthcare facilities, limited access to emergency rooms, and patients’ fear of contracting the SARS-CoV-2 virus, when seeking usual medical help, led to the rapid progression of CVD. The study revealed a reduction of 2% to 5% in dispended lipid-lowering, antidiabetic, and antihypertensive drugs in 2020, compared to the previous year [64]. Further research is needed to assess whether the COVID-19 pandemic will have a long-term impact on mortality due to CVD in the coming years.

Thus, the results obtained in this study confirm the hypothesis about the relationship between air pollution and CVD mortality. We believe that there is a need for further research to better understand both age- as well as sex-related differences in sensitivity to air pollution to identify more efficiently high-risk individuals and protect them in particular in the face of new threats and epidemiological changes associated with COVID-19. Moreover, as Liu et al., showed, air pollution control policies and measures should not be targeted to minimize the influence of a single pollutant but focus on all of them [5].

Several strengths of our study should be noted. First of all, it is a large epidemiological study that has covered 6 million person-years of follow-up. Secondly, the topic of air pollution’s influence on public health was for many years neglected in Eastern Poland and there are not a lot of studies regarding this issue in this region. Thirdly, we analyzed the influence of air pollution not only on the overall population but also depending on people’s age and sex. Lastly, novel findings regarding Polish smog might be especially interesting in the future, when it will be more established in science. Our study showed that even though the air quality in the analyzed region is not as poor as in China or India, it is still harmful and there is no such thing as a safe level of air pollution.

## 5. Limitations

Our study also has several limitations. Firstly, case-specific mortality may have been underestimated due to garbage codes [65]. Secondly, new epidemiological factors, such as the COVID-19 pandemic may have influenced the results in the last year of analysis. Lastly, we used zip codes of residence to connect individual exposure and outcomes which can make for possible misclassifications, due to out-of-data codes or people’s mobility.

## 6. Conclusions

As our study showed, despite progress made in pharmacological treatment and interventional procedures, mortality due to CVD has continued to increase. Therefore, focusing on epidemiological risk factors such as air pollution is now more important than ever.

In our study, a negative effect of PMs was noticeable in both ACS- and IS-related mortality. NO_2_ significantly affected mortality only due to ACS. The detrimental influence of air pollution was the most noticeable the next day after exposure. The most vulnerable to poor air quality groups were women and elderly people. Females were up to 2.5% more sensitive than males, whereas older people were even up to 4.4% more susceptible compared to younger individuals. Weather conditions had no direct effect on ACS- and IS-related mortality. Moreover, we revealed considerable seasonal and weekly variation in the mortality due to ACS and IS.

Systemic changes are crucial to minimizing exposure to smog as there are no safe levels of air pollution. High-risk patients ought to consider undertaking preventive measures. Using a facemask, avoiding walks along congested roads, and monitoring outdoor air pollution levels should all be considered in these individuals.

## Figures and Tables

**Figure 1 ijerph-20-05566-f001:**
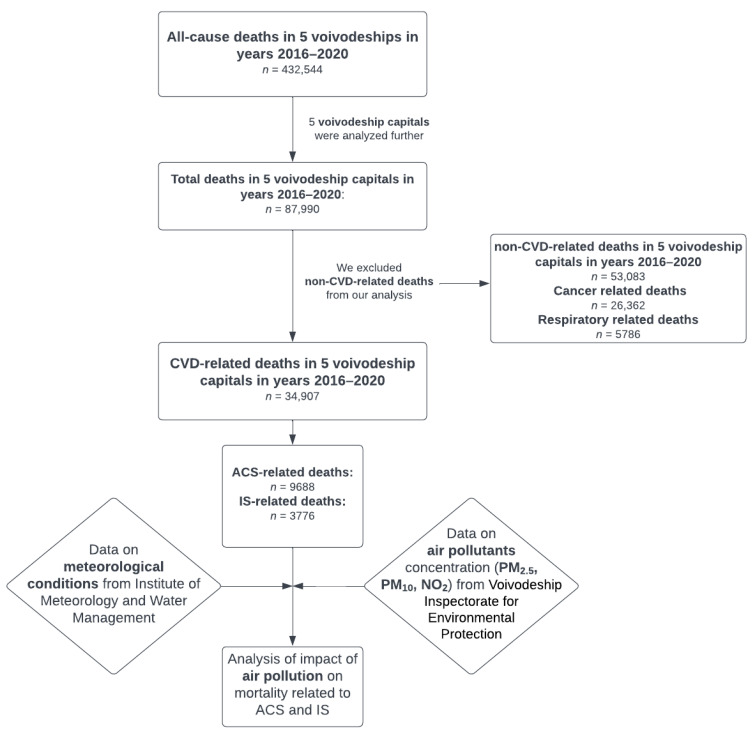
Study design. Abbreviations: ACS, acute coronary syndrome; IS, ischemic stroke; NO_2_, nitrogen dioxide; PM_2.5_, particulate matter with a diameter of 2.5 μm or less; PM_10_ particulate matter with a diameter of 10 μm or less.

**Figure 2 ijerph-20-05566-f002:**
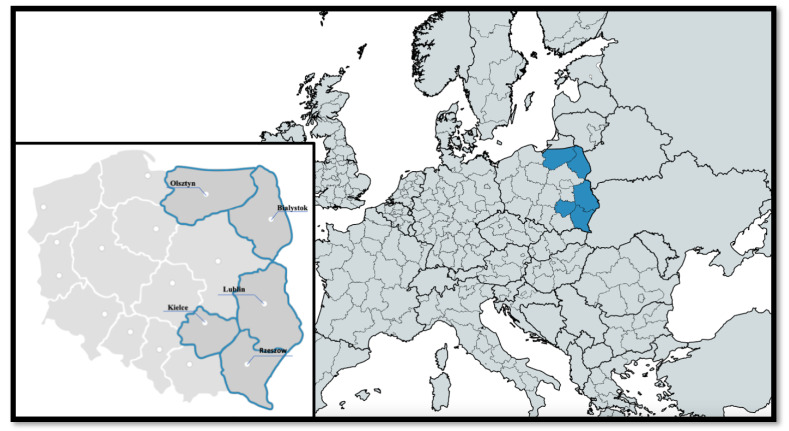
Map of the studied region.

**Figure 3 ijerph-20-05566-f003:**
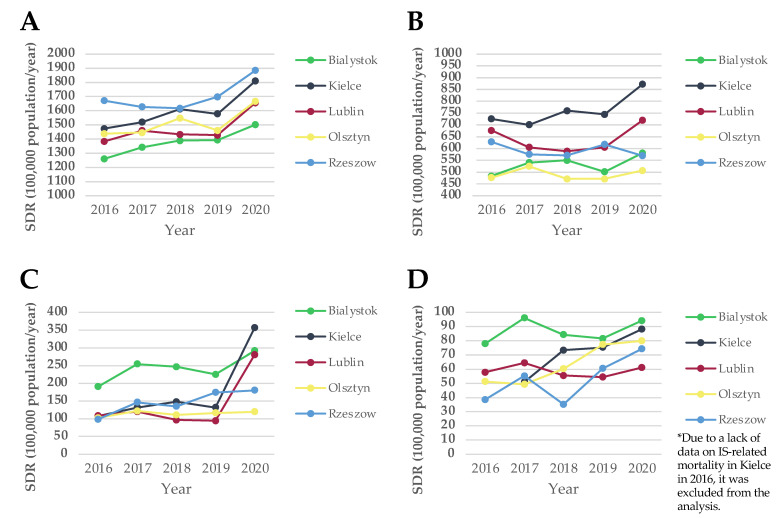
Standardized death rates in analyzed cities in the years 2016–2020. (**A**) all-cause-related SDR, (**B**) CVD-related SDR, (**C**) ACS-related SDR, (**D**) IS-related SDR. Abbreviations: ACS, acute coronary syndrome; CVD, cardiovascular disease; IS, ischemic stroke; SDR, standardized death rate.

**Figure 4 ijerph-20-05566-f004:**
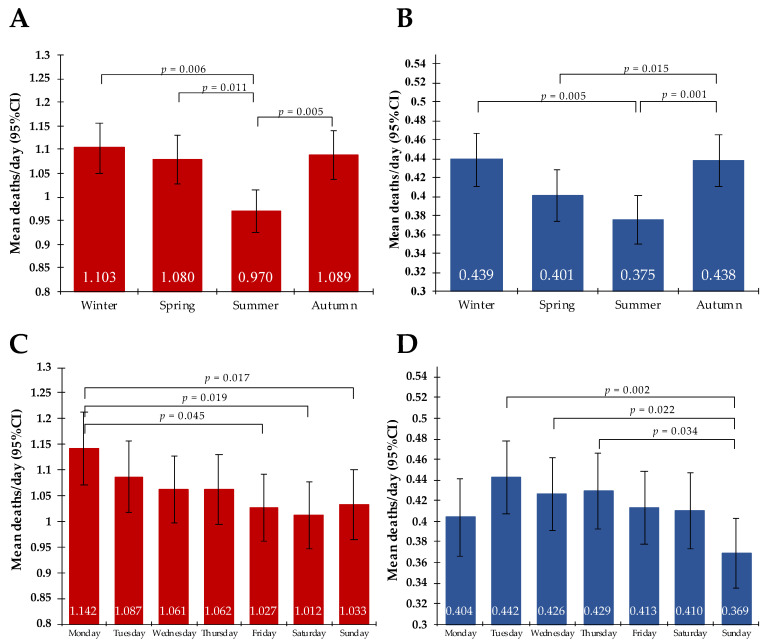
Chronobiological trends of cause-specific mortality. (**A**) a seasonal variation of ACS-related mortality, (**B**) a seasonal variation of IS-related mortality, (**C**) a weekly variation of ACS-related mortality, (**D**) a weekly variation of IS-related mortality. *p*-values are shown for pairwise comparison. Abbreviations: ACS, acute coronary syndrome; CI, confidence interval; IS, ischemic stroke; *p*, *p*-value.

**Figure 5 ijerph-20-05566-f005:**
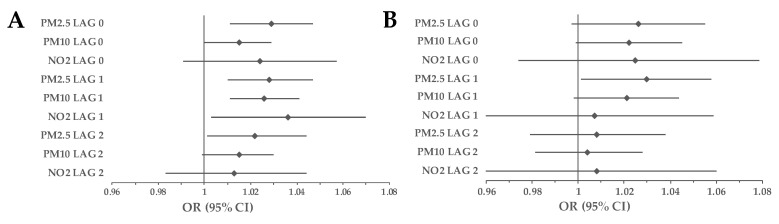
Overall association between exposure to air pollution and cause-specific mortality. (**A**) mortality due to ACS, (**B**) mortality due to IS. Results are presented as ORs with 95% CIs using CLR. Abbreviations: ACS, acute coronary syndrome; CI, confidence interval; CLR, conditional logistic regression; IS, ischemic stroke; NO_2_, nitrogen dioxide; OR, odds ratio; PM_2.5_, particulate matter with a diameter of 2.5 μm or less; PM_10_ particulate matter with a diameter of 10 μm or less.

**Figure 6 ijerph-20-05566-f006:**
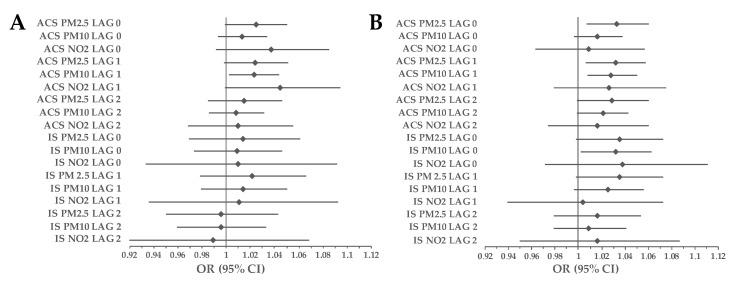
Association between exposure to air pollution and incidence of death related to ACS and IS in the study group separated by sex. (**A**) male population, (**B**) female population. Results are presented as ORs with 95% CIs using CLR. Abbreviations: ACS, acute coronary syndrome; CI, confidence interval; CLR, conditional logistic regression; IS, ischemic stroke; NO_2_, nitrogen dioxide; OR, odds ratio; PM_2.5_, particulate matter with a diameter of 2.5 μm or less; PM_10_ particulate matter with a diameter of 10 μm or less.

**Figure 7 ijerph-20-05566-f007:**
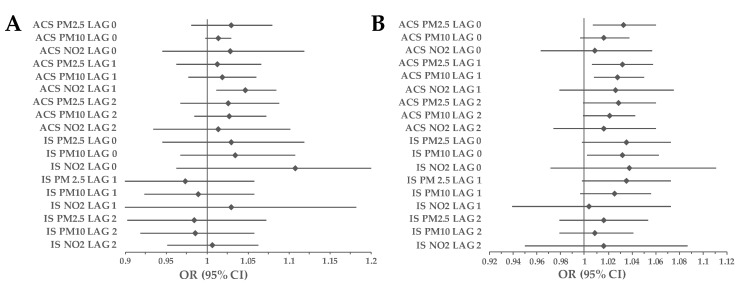
Association between exposure to air pollution and incidence of death related to ACS and IS in the study group separated by age. (**A**) people under 65 years old, (**B**) people over 65 years old. Results are presented as ORs with 95% CIs using CLR. Abbreviations; ACS, acute coronary syndrome; CI, confidence interval; CLR, conditional logistic regression; IS, ischemic stroke; NO_2_, nitrogen dioxide; OR, odds ratio; PM_2.5_, particulate matter with a diameter of 2.5 μm or less; PM_10_ particulate matter with a diameter of 10 μm or less.

**Table 1 ijerph-20-05566-t001:** Characteristics of the studied cities.

City	Bialystok	Kielce	Lublin	Olsztyn	Rzeszow
Inhabitants, *n*	296,958	193,415	338,586	171,249	196,638
Population density, people/km^2^	2908.5	1764.7	2297.1	1938.7	1530.3
Male, %	46.91	46.77	46.13	46.50	47.50
Femininity ratio	113	114	117	115	110
Share of elderly people (>65 years old), %	21.46	25.72	23.13	23.41	21.01

**Table 2 ijerph-20-05566-t002:** Mortality statistics in the five analyzed cities in the years 2016–2020.

Variable	Bialystok	Kielce	Lublin	Olsztyn	Rzeszow	*p*
Total deaths, *n*	19,166	16,390	25,044	12,540	14,850	N/A
2016; N	3543	3002	4699	2366	2902	N/A
2017; N	3769	3106	4953	2378	2832	N/A
2018; N	3900	3302	4872	2571	2830	N/A
2019; N	3743	3251	4856	2451	2991	N/A
2020; N	4211	3729	5664	2774	3295	N/A
Male. N (%)	9797 (51.12)	8590 (52.41)	12751 (50.91)	6641 (52.96)	7871 (53)	<0.001
Mean age (SD)	73.86 (15.94)	72.7 (15.71)	72.64 (16.8)	71.83 (16.8)	72.02 (17.84)	<0.001
CDR, (100,000 population/year)	1290.82	1694.8	1479.33	1464.53	1510.49	<0.001
SDR, (100,000 population/year)	1365.84	1598.27	1471.91	1511.87	1699.7	<0.001
CVD-related deaths, *n* (% of total deaths)	7293 (38.05)	7729 (47.16)	10,764 (42.98)	4039 (32.21)	5082 (34.22)	<0.001
2016; N (% of total deaths)	1330 (37.54)	1464 (48.77)	2275 (48.41)	773 (32.67)	1074 (37.01)	<0.001
2017; N (% of total deaths)	1484 (39.37)	1422 (45.78)	2028 (40.94)	861 (36.21)	983 (34.71)	<0.001
2018; N (% of total deaths)	1507 (38.64)	1544 (46.76)	1988 (40.8)	779 (30.3)	976 (34.49)	<0.001
2019; N (% of total deaths)	1376 (36.76)	1517 (46.66)	2040 (42.01)	787 (32.11)	1067 (35.67)	<0.001
2020; N (% of total deaths)	1596 (37.9)	1782 (47.79)	2433 (42.96)	839 (30.25)	982 (29.8)	<0.001
Male, N (%)	3262 (44.73)	3774 (48.83)	5073 (47.13)	2096 (51.89)	2462 (48.45)	<0.001
Mean age (SD)	79.94 (11.32)	77.35 (12.72)	77.91 (12.7)	75.7 (13.38)	78.01 (12.88)	<0.001
CVD CDR (100,000 population/year)	491.18	799.21	635.82	471.71	516.89	<0.001
CVD SDR (100,000 population/year)	531.23	760.4	638.92	490.18	592.16	<0.001
ACS-related deaths, *n* (% of CVD deaths)	3299 (45.24)	1793 (23.2)	2379 (22.1)	963 (23.84)	1254 (24.68)	<0.001
Male, N (%)	1591 (48.23)	921 (51.37)	1212 (50.95)	602 (62.51)	626 (49.92)	<0.001
Mean age (SD)	79.43 (9.66)	77.6 (12.42)	77.8 (11.94)	73.55 (11.34)	79.05 (11.55)	<0.001
ACS CDR (100,000 population/year)	222.19	185.4	140.53	112.47	127.54	<0.001
ACS SDR (100,000 population/year)	241.9	175.5	140.23	114.56	147.12	<0.001
IS-related deaths, *n* (% of CVD deaths)	1188 (16.29)	624 (8.07)	986 (9.16)	527 (13.05)	451 (8.87)	<0.001
Male, N (%)	487 (40.99)	295 (47.28)	417 (42.29)	247 (46.87)	194 (43.02)	<0.001
Mean age (SD)	80.67 (10.06)	77.81 (11.29)	79.62 (11.17)	77.22 (11.65)	78.1 (11.48)	0.046
IS CDR (100,000 population/year)	80.01	64.52	58.24	61.55	45.87	<0.001
IS SDR (100,000 population/year)	86.79	60.61	58.71	63.61	52.79	<0.001

Abbreviations: ACS, acute coronary syndrome, CDR, crude death rate; CVD, cardiovascular disease; IS, ischemic stroke; N/A, not applicable; SD, standard deviation; SDR, standardized death rate.

**Table 3 ijerph-20-05566-t003:** Statistics on air pollution data.

City	Bialystok	Kielce	Lublin	Olsztyn	Rzeszow
Air pollutant	PM_2.5_				
Days with observation; *n* (%)	1778 (97.32)	1784 (97.65)	1789 (97.92)	1772 (96.99)	1784 (97.65)
2016; mean/day (SD)	19 (12.66)	23.39 (23.39)	26.53 (17.39)	15.25 (10.11)	22.21 (14.28)
2017; mean/day (SD)	16.75 (14.12)	24.76 (24.76)	22.01 (20.12)	16.82 (13.3)	24.13 (23.26)
2018; mean/day (SD)	16.35 (11.67)	24.93 (24.93)	24.38 (18.16)	19.56 (13.21)	22.85 (16.96)
2019; mean/day (SD)	13.42 (8.59)	19.94 (19.94)	20.31 (12.92)	15 (8.96)	19.85 (15.11)
2020; mean/day (SD)	13.46 (8.85)	16.46 (16.46)	18.69 (12.6)	13.96 (8.63)	20.38 (12.67)
Total; mean/day (SD)	15.82 (11.59)	21.84 (21.84)	22.35 (16.73)	16.11 (11.19)	21.88 (16.92)
Daily median	12.38	15.53	17.41	12.7	16.7
Winter season; mean/day (SD)	24.38 (15.76)	36.18 (29.26)	33.7 (22.79)	23.07 (14.76)	33.27 (23.73)
Spring season; mean/day (SD)	11.13 (4.82)	13.92 (6.97)	14.54 (7.24)	11.21 (5.16)	15.1 (6.39)
Summer season; mean/day (SD)	9.15 (4.44)	12.21 (6.11)	13.93 (6.33)	10.31 (4.47)	13.77 (6.18)
Autumn season; mean/day (SD)	19.02 (10.71)	25.39 (16.64)	27.21 (15.62)	20.47 (11.35)	25.91 (16.65)
Exceeded daily mean AQG 2021; *n* (%)	666 (37.46)	924 (51.79)	1054 (58.92)	708 (39.95)	1016 (56.95)
Air pollutant	PM_10_				
Days with observation; *n* (%)	1732 (94.8)	1776 (97.21)	1813 (99.23)	1787 (97.81)	1821 (99.67)
2016; mean/day (SD)	23.98 (13.51)	34.87 (34.87)	30.92 (18.85)	24.41 (12.98)	27.32 (15.32)
2017; mean/day (SD)	23.25 (15.71)	36.38 (36.38)	32.47 (23.53)	23.39 (15.53)	30.33 (25.45)
2018; mean/day (SD)	26.35 (15.61)	37.41 (37.41)	33.59 (20.8)	26.47 (15.77)	31.19 (18.36)
2019; mean/day (SD)	21.13 (11.22)	31.93 (31.93)	26.47 (14.65)	19.93 (11.1)	24.41 (17.68)
2020; mean/day (SD)	21.49 (12.86)	24.89 (24.89)	22.35 (13.46)	18.08 (9.96)	19.86 (10.56)
Total; mean/day (SD)	23.17 (13.96)	32.99 (32.99)	29.18 (19.11)	22.43 (13.58)	26.62 (18.57)
Daily median	19.93	26.48	24.28	19.3	21.02
Winter season; mean/day (SD)	28.77 (18.49)	47.39 (33.69)	41.3 (26.84)	28.33 (16.63)	38.27 (26.64)
Spring season; mean/day (SD)	20.09 (8.39)	26.7 (12.06)	23.69 (11.15)	19.63 (10.89)	21.43 (8.55)
Summer season; mean/day (SD)	18.58 (8.09)	23.66 (9.74)	20.87 (9.12)	17.33 (9.1)	18.44 (7.03)
Autumn season; mean/day (SD)	24.94 (15.31)	34.55 (20.29)	31.42 (17.88)	24.58 (13.89)	29.07 (18.72)
Exceeded daily mean AQG 2021; *n* (%)	120 (6.93)	347 (19.54)	251 (13.84)	116 (6.49)	203 (11.15)
Air pollutant	NO_2_				
Days with observation; *n* (%)	1799 (98.47)	1788 (97.87)	1826 (99.95)	1818 (99.51)	1827 (100)
2016; mean/day (SD)	13.46 (5.55)	25.66 (11.09)	21.74 (9.37)	15.26 (6.41)	18.65 (6.52)
2017; mean/day (SD)	13.09 (5.19)	23.26 (13.04)	21.73 (9.71)	14.1 (8.66)	17.59 (8.55)
2018; mean/day (SD)	14.05 (6.26)	26.79 (10.4)	21.63 (10.07)	14.72 (9.53)	18.06 (7.75)
2019; mean/day (SD)	12.55 (5.2)	21 (7.94)	19.49 (8.45)	12.7 (6.86)	15.78 (7.3)
2020; mean/day (SD)	11.93 (5.53)	24.36 (9.15)	17.2 (7.05)	11.47 (6.57)	12.25 (5.35)
Total; mean/day (SD)	13.03 (5.6)	24.16 (10.64)	20.36 (9.16)	13.65 (7.83)	16.47 (7.54)
Daily median	11.89	22.48	18.68	12.38	15.02
Winter season; mean/day (SD)	14.55 (6.26)	28.76 (12.15)	23.24 (10.32)	16.83 (8.46)	20.61 (9.23)
Spring season; mean/day (SD)	11.74 (4.59)	20.87 (8.86)	17.98 (7.23)	9.99 (5.85)	14.11 (5.38)
Summer season; mean/day (SD)	12.42 (4.92)	22.92 (9.97)	19.11 (8.79)	11.28 (5.61)	13.61 (4.92)
Autumn season; mean/day (SD)	13.46 (6.06)	24.2 (9.9)	21.26 (9.29)	16.55 (8.42)	17.8 (7.73)
Exceeded daily mean AQG 2021; *n* (%)	69 (3.83)	740 (41.39)	491 (26.89)	141 (7.76)	228 (12.48)

Abbreviations; AQG, Air Quality Guidelines; NO_2_, nitrogen dioxide; PM_2.5_, particulate matter with a diameter of 2.5 μm or less; PM_10_ particulate matter with a diameter of 10 μm or less; SD, standard deviation.

## Data Availability

The data that support the findings of this study are available from the corresponding author on request.

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
