# Peer review of "Multi-City Analysis of the Acute Effect of Polish Smog on Cause-Specific Mortality (EP-PARTICLES Study)"

_ijerph, 2023, doi:10.3390/ijerph20085566_

Round 1

Reviewer 1 Report (Previous Reviewer 1)

Many thanks for taking time out to improve the content. Happy to recommend the work be considered in its present format.   

Author Response

We would like to thank you very much for taking the time to improve our manuscript.

Reviewer 2 Report (Previous Reviewer 3)

The following few comments are not well modified:

1. introduction, In this section, the summary of the current research results is very lacking, and more adjustments and revisions are needed to supplement more relevant research results. It is necessary to supplement some researches in recent years:

https://doi.org/10.1016/j.envres.2022.114095

https://doi.org/10.1016/j.envpol.2021.116770

https://doi.org/10.1016/j.envres.2020.110290

https://doi.org/10.1016/j.apr.2021.101247

2. Results, It is necessary to carry out a detailed analysis in combination with your own data, and give more quantitative results. In addition, the explanation for the cause of the phenomenon must be based on the basis and specific data, rather than arbitrary explanations.

3. The quality of the figures in the article is very poor and needs to be redrawn, especially figures 3-7.

4. Conclusion. need to be re-written, needs to focus on giving quantitative results. It’s too short now.

Author Response

The following few comments are not well modified:

Point 1: Introduction, In this section, the summary of the current research results is very lacking, and more adjustments and revisions are needed to supplement more relevant research results. It is necessary to supplement some researches in recent years:

https://doi.org/10.1016/j.envres.2022.114095

https://doi.org/10.1016/j.envpol.2021.116770

https://doi.org/10.1016/j.envres.2020.110290

https://doi.org/10.1016/j.apr.2021.101247

Response 1: We would like to thank the Reviewer for the effort to improve our manuscript's quality. We implemented the papers suggested by you into our introduction sections to further improve the quality of this section. Please notice changes to the current introduction.

Point 2: Results, it is necessary to carry out a detailed analysis in combination with your own data, and give more quantitative results. In addition, the explanation for the cause of the phenomenon must be based on the basis and specific data, rather than arbitrary explanations.

Response 2: We are attaching a more detailed analysis of the influence of weather conditions such as temperature, humidity, and atmospheric pressure in supplementary appendix, as we feel our results section is already too extensive. Moreover, we added a comparison between the effects of air pollution on both sexes and age groups. Please notice changes to our supplementary materials and results section.

Point 3: The quality of the figures in the article is very poor and needs to be redrawn, especially figures 3-7.

Response 3: We replaced old figures with new ones in high definition. The quality and visibility of the figures have significantly improved.

Point 4: Conclusion. need to be re-written, needs to focus on giving quantitative results. It’s too short now.

Response 4: We further improved the quality of the conclusion sections; however, our goal is to keep it as concise as possible.

Reviewer 3 Report (New Reviewer)

I only have a few minor comments (as detailed below). However, a more major comment is regarding to the impact of temperature and relative humidity. These were included in the analysis but there was no mention in the results and discussion sections. I believe it is worth including these results.

1.      Abstract: mention that the “lag” is in days.

2.      Figure 4: In figure 4 and in text “p” is used. This is a bit confusing. Somewhere it should be mentioned that “p” is used for the “p-value”.

3.      Figure 6: In the caption the term “in the study group divided by sex” is used. It might be better as “in the study groups separated by sex”. As when I first read it, I thought that the “divided” was used to imply division (as in x/y).

4.      Figures 5, 6 and 7. It might be more informative to the reader if it is specified from what test the results were obtained. In this was the reader does not have to go back to the methods section. Also, the x-axis labels, were the “,” mean to be “.”, e.g., is “0,1” mean to be “0.1” etc.

Author Response

Point 1: I only have a few minor comments (as detailed below). However, a more major comment is regarding to the impact of temperature and relative humidity. These were included in the analysis but there was no mention in the results and discussion sections. I believe it is worth including these results.

Response 1: Thank you for your time, effort, and your kind comments. We have attached an analysis of weather conditions (such as temperature, relative humidity, or atmospheric pressure) impact on mortality due to ischemic stroke and acute coronary syndromes in supplementary appendix. There was no statistically significant influence of the aforementioned factors on the number of deaths. Please notice changes to the supplementary material.

Table 1. Association between weather conditions and occurrence of ACS- and IS-related deaths. Results are presented as ORs with 95% CIs using CLR.

Mortality

LAG

Group

Mean temperature

Mean humidity

Mean atmospheric pressure (sea level)

OR

95% CI

p

OR

95% CI

p

OR

95% CI

p

ACS

LAG 0

Overall population

1.021

0.956-1.089

0.54

0.986

0.962-1.011

0.28

0.981

0.951-1.011

0.21

Male

1.03

0.94-1.128

0.53

0.972

0.939-1.006

0.11

0.982

0.94-1.025

0.4

Female

1.012

0.923-1.109

0.81

1.002

0.966-1.038

0.93

0.98

0.937-1.024

0.36

Over 65 years old

1.029

0.959-1.104

0.43

0.994

0.967-1.021

0.65

0.981

0.949-1.014

0.25

Under 65 years old

0.971

0.819-1.151

0.73

0.945

0.885-1.008

0.08

0.979

0.902-1.063

0.62

LAG 1

Overall population

0.986

0.923-1.054

0.68

0.998

0.973-1.024

0.88

0.985

0.955-1.016

0.35

Male

0.971

0.885-1.066

0.54

0.988

0.954-1.024

0.52

1

0.957-1.045

0.99

Female

1.002

0.913-1.101

0.96

1.009

0.972-1.046

0.65

0.971

0.929-1.015

0.19

Over 65 years old

0.986

0.918-1.059

0.7

1.007

0.98-1.035

0.61

0.986

0.953-1.02

0.41

Under 65 years old

0.983

0.826-1.17

0.85

0.947

0.886-1.011

0.1

0.979

0.902-1.063

0.61

LAG 2

Overall population

0.978

0.917-1.043

0.49

0.991

0.966-1.016

0.47

0.989

0.963-1.015

0.4

Male

0.974

0.889-1.066

0.57

0.984

0.949-1.02

0.37

0.994

0.959-1.031

0.76

Female

0.982

0.896-1.076

0.69

0.998

0.962-1.035

0.92

0.983

0.947-1.02

0.37

Over 65 years old

0.975

0.909-1.045

0.47

0.994

0.967-1.022

0.67

0.984

0.957-1.012

0.26

Under 65 years old

0.994

0.838-1.179

0.95

0.971

0.907-1.04

0.4

1.017

0.95-1.09

0.62

IS

LAG 0

Overall population

1.021

0.956-1.089

0.54

0.986

0.962-1.011

0.28

0.981

0.951-1.011

0.21

Male

1.03

0.94-1.128

0.53

0.972

0.939-1.006

0.11

0.982

0.94-1.025

0.4

Female

1.012

0.923-1.109

0.81

1.002

0.966-1.038

0.93

0.98

0.937-1.024

0.36

Over 65 years old

1.029

0.959-1.104

0.43

0.994

0.967-1.021

0.65

0.981

0.949-1.014

0.25

Under 65 years old

0.971

0.819-1.151

0.73

0.945

0.885-1.008

0.08

0.979

0.902-1.063

0.62

LAG 1

Overall population

0.986

0.923-1.054

0.68

0.998

0.973-1.024

0.88

0.985

0.955-1.016

0.35

Male

0.971

0.885-1.066

0.54

0.988

0.954-1.024

0.52

1

0.957-1.045

0.99

Female

1.002

0.913-1.101

0.96

1.009

0.972-1.046

0.65

0.971

0.929-1.015

0.19

Over 65 years old

0.986

0.918-1.059

0.7

1.007

0.98-1.035

0.61

0.986

0.953-1.02

0.41

Under 65 years old

0.983

0.826-1.17

0.85

0.947

0.886-1.011

0.1

0.979

0.902-1.063

0.61

LAG 2

Overall population

0.978

0.917-1.043

0.49

0.991

0.966-1.016

0.47

0.989

0.963-1.015

0.4

Male

0.974

0.889-1.066

0.57

0.984

0.949-1.02

0.37

0.994

0.959-1.031

0.76

Female

0.982

0.896-1.076

0.69

0.998

0.962-1.035

0.92

0.983

0.947-1.02

0.37

Over 65 years old

0.975

0.909-1.045

0.47

0.994

0.967-1.022

0.67

0.984

0.957-1.012

0.26

Under 65 years old

0.994

0.838-1.179

0.95

0.971

0.907-1.04

0.4

1.017

0.95-1.09

0.62

Abbreviations: ACS, acute coronary syndrome; CI, confidence interval; CLR, conditional logistic regression; IS, ischemic stroke; OR; odds ratio; p, p-value

Point 2: Abstract: mention that the “lag” is in days.

Response 2: Thank you for noticing, we have modified the abstract accordingly.

Point 3: Figure 4: In figure 4 and in text “p” is used. This is a bit confusing. Somewhere it should be mentioned that “p” is used for the “p-value”.

Response 3: We agree that it may be confusing for potential readers. We have added an explanation that “p” stands for “p-value” in both text and tables.

Point 4: Figure 6: In the caption the term “in the study group divided by sex” is used. It might be better as “in the study groups separated by sex”. As when I first read it, I thought that the “divided” was used to imply division (as in x/y).

Response 4: We have replaced “divided” for “separated” in Figure 6 and Figure 7.

Point 5: Figures 5, 6 and 7. It might be more informative to the reader if it is specified from what test the results were obtained. In this was the reader does not have to go back to the methods section. Also, the x-axis labels, were the “,” mean to be “.”, e.g., is “0,1” mean to be “0.1” etc.

Response 5: Thank you for your suggestion. We have corrected the x-axis labels and replaced “,” with “.”. Moreover, we have added what test we used in the captions of Figures 5-7.

Round 2

Reviewer 2 Report (Previous Reviewer 3)

accept

This manuscript is a resubmission of an earlier submission. The following is a list of the peer review reports and author responses from that submission.

Round 1

Reviewer 1 Report

Many  thanks for the chance to review your work. Please consider the following top help you improve the work quality: 

i. To ensure clarity and assimilation of its content with ease, consider rewriting the abstract providing detailed account of work done and its implication. The use of figure to inform this section has greatly impact on the content. 

ii. This sentence "Even though indoor air pollution has become less of a problem in the last thirty years with a 70% reduction in deaths per 100,000 people worldwide, outdoor air pollution remains a serious threat to public health, even in high-income countries such as Poland" need further justification to strengthen the argument contain here. Considering that most individual spend their time indoor exposed to pollution with indoor and outdoor origin and part of the discussion section in your work stated "When analysing age-specific differences in air pollution impact on ACS- and IS-related mortality, a more noticeable effect is present in elderly people" you will agree it will be hard not to provide further evidence to support what was stated herein.   

iii. Generally, the introduction section will need further development to help strengthen the case for the work considered in the article. In its present it is hard to see the gap contained here that the study aim to address. 

Reviewer 2 Report

It is well known and largely established and confirmed herewith that short-term exposure to air pollution due to particulate matter (PM2.5 and PM10) as well as to a gaseous component as NO2 is associated with a short-term impact on mortality ad morbidity due to atherothrombotic cardiovascular diseases, such as the acute coronary syndrome (ACS) and ischemic stroke. It is also well known that Eastern Poland is a European area characterized by high levels of air pollution. In this study the authors chose to evaluate over a 4-year period (2016-2020) whether or not the well-established association between air pollution and atherothrombotic CVD was also confirmed in this highly polluted European region. The authors found that a short-term 10 µg/mm3 increase of the three main pollutants (PM2.5, PM10 and NO2) was associated with an increased mortality due to ACS and ischemic stroke. An additional finding was that the negative influence of particulate matter was particularly prominent in women and, not surprisingly, in older people. 

On the whole, this reviewer found that this study was correctly done and written. The findings are solid and epidemiologically plausible but not novel at all, nor of general scientific interest beyond their impact in Poland.  Hence, this study and the related manuscript can be defined as confirmation of already available knowledge with no advance of the general knowledge.

Reviewer 3 Report

Multi-city analysis of an acute effect of air pollution on cause specific mortality in Eastern Poland (EP-PARTICLES study). The topic and some findings are interesting. This manuscript not only provides some phenomena, but also is a good job of explaining the mechanism. In all, the manuscript can be published after minor revision. The specific comments are listed as bellow:

1. There are many spelling and formatting errors in the article, which need to be revised carefully.

2. introduction, In this section, the summary of the current research results is very lacking, and more adjustments and revisions are needed to supplement more relevant research results. It is necessary to supplement some researches in recent years:

https://doi.org/10.1016/j.chemosphere.2020.127884

https://doi.org/10.1016/j.envres.2022.114095

https://doi.org/10.1016/j.envpol.2021.116770

https://doi.org/10.1016/j.envres.2020.110290

https://doi.org/10.1016/j.apr.2021.101247

https://doi.org/10.1016/S2542-5196(20)30272-2

3. Results, It is necessary to carry out a detailed analysis in combination with your own data, and give more quantitative results. In addition, the explanation for the cause of the phenomenon must be based on the basis and specific data, rather than arbitrary explanations.

4. The quality of the figures in the article is very poor and needs to be redrawn, especially figures 3-7.

5. Conclusion. need to be re-written, needs to focus on giving quantitative results. It’s too short now.

Round 2

Reviewer 2 Report

The manuscript had to be rejected for lack of any novelty. Formally it was already ok .